# Knowledge Repatriation: A Pilot Project about Making Cedar Root Baskets

Sharon M. Fortney

Museum of Vancouver, 1100 Chestnut Street, Vancouver, BC V6J 3J9, Canada; sfortney@museumofvancouver.ca

**Abstract:** This paper describes the first phase of a Coast Salish Knowledge Repatriation Project being coordinated by the Curator of Indigenous Collections and Engagement at the Museum of Vancouver, within the unceded, ancestral territories of the xʷməθkʷəy̓əm (Musqueam), Sḵwx̱wú7mesh (Squamish), and səlilwətaɬ (Tsleil-Waututh) nations. The goal of this knowledge repatriation work is to support cultural revitalization and language renewal through activities that generate learning opportunities for community members. These activities pivot around knowledge that has been lost due to urbanization, forced assimilation efforts, and other colonial activities that may have restricted access to traditional lands and resources, preventing knowledge transmission. This work is about shifting the focus from extractive projects, that benefit external audiences, to one that supports capacity building and cultural renewal within communities. This essay describes a project to reintroduce coiled cedar root basketry into communities within the Greater Vancouver area in the province of British Columbia, Canada.

**Keywords:** knowledge repatriation; Coast Salish; coiled basketry; reconciliation; redress; traditional knowledge

## 1. Introduction

In the Canadian museum community, the term repatriation is associated with the return of belongings, or ancestral remains, to communities or families of origin. It is an action taken to rectify the wrongdoing that occurred as a result of colonialism, during eras when the Potlatch Ban and Residential School Act were actively enforced in Canada and many public and private collections were being formed.[1] Repatriation work may also occur with intellectual property when it has been documented through sound recordings, archival documents, or photography. Source communities typically undertake this type of work with museums and other repositories such as archives. Historically, the onus has been on Indigenous communities to locate their belongings[2] and intellectual property, and petition for their return, at their own expense. Repatriation work always requires an investment of time, capacity, and funding to carry a project from concept to completion. This burden has rested with Indigenous communities as there are few funding opportunities in Canada to support this work, and they have to date been sporadic in nature.

In this paper, I propose another form of repatriation which I call "Knowledge Repatriation". The goal of this type of work is to support cultural revitalization and language renewal through activities that generate learning opportunities for community members. These activities pivot around knowledge that has been lost due to urbanization, forced assimilation efforts, and other colonial activities that may have restricted access to traditional lands and resources, preventing knowledge transmission.

The goal of "Knowledge Repatriation" is to use museum time and resources to fundraise, project manage, and document the resulting learning opportunities as a way of supporting capacity within participating Indigenous communities. The objective is to shift the work of the museums away from extractive projects, that primarily serve the needs of external audiences such as tourists and non-Indigenous visitors, to those that support

the self-identified needs of communities. It is not just about recognizing these community priorities, but also seasonal cycles and community-based timelines. This paper will describe the first phase of a Coast Salish Knowledge Repatriation Project being coordinated by the Curator of Indigenous Collections and Engagement at the Museum of Vancouver, within the unceded, ancestral territories of the xʷməθkʷəýəm (Musqueam), Sḵwx̱wú7mesh (Squamish), and səlilwətaɬ (Tsleil-Waututh) nations.

## 2. Recognizing the Extractive Nature of Many Reconciliation Efforts

For many years I worked as a contract researcher, writer, and guest curator in the Greater Vancouver area on projects that led me to connect with friends and colleagues working for the host nations of xʷməθkʷəýəm (Musqueam), Sḵwx̱wú7mesh (Squamish), and səlilwətaɬ (Tsleil-Waututh). On one occasion, in 2017, I had three distinct projects to discuss with a friend from the Language and Culture team at Squamish Nation—each on behalf of a different museum. This really drove home to me the number of requests that these nations were fielding on a daily basis from a diversity of institutions and businesses within the Greater Vancouver area.[3] Although these efforts are meant to be respectful, often those involved do not understand the nature of their requests and the amount of work that may go into meeting each request. Over the years, as the desire has grown within the region to prioritize activities that are of a reconciliatory nature, there has also been an increasing interest in accessing Indigenous languages for different types of initiatives, among them naming and commemorating places within the city. An example is the renaming of two civic plazas in 2018—šxʷƛ̓exən Xwtl'a7shn (formerly known as the Queen Elizabeth Theatre Plaza) and šxʷƛ̓ənəq Xwtl'e7énḵ Square (formerly the Vancouver Art Gallery north plaza).[4]

Each of these places was given a name in each of the two local Salish languages (hən̓q̓əmin̓əm̓ and Sḵwx̱wú7mesh sníchim). These are both endangered languages. This project celebrates the ongoing presence of the three host nations in their traditional lands. To ensure that the project was carried out respectfully, language speakers from each community had to seek advice from their respective elders and knowledge holders before they could work together to create these contemporary place names.[5] It required an investment of time and resources to accomplish this task. I would argue that while this type of commemorative work is important, the final result is situated within an urban setting and the audience is largely external. Working with language team members, and other cultural staff from Vancouver's host nations, I have heard several express that their personal priorities are inward-focused. There is a desire to create curriculum materials for teaching their endangered languages to community members to increase fluency within the community. They are excited about immersive learning opportunities, such as the launch of a "Ta Tsíptsipi7lhkn (Language Nest)" by the Ta na wa Ns7éyx̱nitm ta Snew̓íyelh department of the Squamish Nation, or the creation of hən̓q̓əmin̓əm̓ illustrated storybooks as was carried out by the Musqueam Language team for the c̓əsnaʔəm exhibition project.[6] These types of learning opportunities directly benefit members of their respective communities.

While we undertook our work together for the long-term exhibition, "That Which Sustains Us," which opened at MOV in 2020, the host nation representatives who formed our curatorial collective expressed a desire to commission a cedar root basket for a section that dealt with forests and food sovereignty. One of the Musqueam representatives suggested that the process be filmed, and Indigenous languages be added to describe the process. Coiled cedar root baskets are quintessentially Salish belongings, found in many homes, both Indigenous and non-Indigenous, and often in abundance in museum collections in Canada. They come in many shapes and sizes, some styles specific to certain types of work and gathering activities, while others are unique items made for collectors—teacups, tables, and violin cases being some examples. This abundance has led many to believe that these baskets are strictly utilitarian in nature, yet when you examine how the knowledge of their production is transmitted on the coast (within families), and some of the ceremonial and

political contexts in which they are used, it becomes obvious that root weavers were highly esteemed specialists in the Salish world (Fortney 2000, 2001, 2022).

It was not possible to complete the basketry project as part of "That Which Sustains Us," as we encountered several capacity issues—many stemming from the impacts of the COVID-19 pandemic, which created barriers to gathering in person. However, this idea was set aside but not forgotten. At the next meeting of our community curatorial collective, several months after the opening of "That Which Sustains Us," we discussed shifting the focus of our work together. In the years prior, we reacted to exhibition projects that had been put in place by staff prior to my employment at the museum. (The planning process for exhibits at MOV is three years, as we require time to fundraise and undertake meaningful community engagement before we can move towards design and fabrication.) Rather than starting with the idea of a new exhibition project, to serve the needs of external audiences, we decided to carry out work that was of direct benefit to the communities. By carrying out work that could be used for language and culture renewal, to create curriculum resources, we shift the narrative away from extraction to one of capacity building with the communities. The idea was to complete several of these types of projects, beginning with cedar root basketry. There has already been an expressed desire to share this work with a wider audience, and several participants see it as a model for relationship building as it prioritizes reparation between the museum and the host nations. This is significant as the museum sits upon an ancestral village site, illegally annexed by the city.

In the remainder of this essay, I will discuss how we undertook the work of knowledge repatriation to create coiled cedar root baskets at the Museum of Vancouver. I view this as the first phase of a longer-term partnership between our four communities—xʷməθkʷə́ýəm (Musqueam), Sḵwx̱wú7mesh (Squamish), and səlilwətaɬ (Tsleil-Waututh) and the Museum of Vancouver. This work is intended as a case study within a growing body of literature about decolonizing museums and cultural work with Indigenous communities (Ames 1992; McMaster 1992; Peers and Brown 2003; Lonestree 2012; Duffek et al. 2021; Bunn-Marcuse and Jonaitis 2022; Shelton et al. 2022).

## 3. How Long Does It Take to Make a Basket?

I come from a basket making family. My grandmother Mary Ann Pollner, and her mother Annie Chapman, from the Klahoose Nation, both carried this knowledge. Our lineage connects to the Pielle and Timothy families (Annie's parents were Billy Pielle and Martha Timothy).[7] I inherited my grandmother's personal basket collection, made by my great grandmother as gifts at the time of her oldest daughter's marriage in 1937. As a teen, my grandmother offered to share this knowledge with me if I would "go get some roots". Growing up in an urban environment, I really did not know where to start. I was also very allergic to pollen, which was an impediment to harvesting in the spring. In my late twenties, as I was planning to move back to British Columbia from Alberta, we were once again discussing making baskets. The death of my grandmother, months prior to my return, prevented this from happening. Instead, I inherited her awl and the family baskets, and I began to study basketry with elders and other knowledge holders from the Fraser and Squamish valleys as a graduate student at UBC. As a museum worker, I have had many opportunities to study and talk about baskets with community members over the years. I also attended the very first "Fraser Canyon Roots Workshop" in Yale, organized by the late Irene Bjerky who was a descendant of Spuzzum basket makers.[8] All of these experiences guided me in my approach to curating the Knowledge Repatriation Project.

Museum visitors will often look at a basket and say, "How long does it take to make that?" I find such questions difficult to answer as there are so many variables. Do we include harvesting time? Are we considering the skill level of the person who made it? Do we assume that the person simply sat and made the basket and was not interrupted by other demands of daily life? Does this question really tell us anything important about basket making? Not really.

Cedar root basketry can tell us a lot about where people go in their territory at different times of the year. It involves not just the gathering of roots, but the decorative materials and those needed for the foundations of the basket. Gathering is not just about harvesting; it is about preparing materials for use, and this may take time. In my mind, organizing an effective Knowledge Repatriation project required first learning about harvesting the necessary materials. It is not a one-day event, but several events, some of them requiring more than one day. We also needed to follow the seasonal cycles that guide this work, beginning our harvesting in the spring, but setting aside the work in summer when our teacher was at fish camp, and resuming it in the winter when there was more time.

Basket making is a highly personal enterprise as most basket makers learn from other family members, and their teachings, designs, and personal preferences can vary (Bjerky 2006). In some instances, basket makers have been unwilling to teach root basketry outside of their families (Miller 2007, p. 19). Historically, it would also seem that Interior Salish communities had more root weavers than Coast Salish ones. Some older basket makers have suggested that the tradition moved from the interior to the coast, among them Rose Mitchell of Klahoose (Kennedy and Bouchard 1983, p. 76). Interior Salish communities like Mount Currie and Spuzzum have reported basket makers in most families (Laforet and York 1998, p. 97; Wilson 1964), but this has not been the case when I have spoken to basket makers from Coast Salish communities.

For our project we needed to locate a knowledge holder that would be willing to share their knowledge with members of other communities, and consent to do so on film. One of my colleagues at Squamish Nation had begun to learn about cedar roots from affinal relations from Mount Currie, where the Lil'wat reside. She felt that her teacher, Gabrielle (Gay) Williams, would be willing to teach others for educational purposes. She had also been featured in documentary films previously (Obomsawin 2009). After a very thorough discussion about the project, and how the films would be used and accessed, Gay consented to be our teacher. We were very fortunate to have her work with us, as she was not a solo act, but at times was accompanied by her mother, sister, daughter, and grandchildren. Her daughter Pilasi Kingfisher was a constant presence, assisting us with finding the correct harvesting locations and lending her hands as we practiced the various skills that Gay was teaching.

One of the hardest tasks for this project was planning our harvesting excursions in advance, as many things are dependent upon the growth cycles of the plants and specific types of weather conditions. We also needed enough advance notice to ensure that we would have participants from each nation at each event and to secure harvesting sites within the city for each resource. To accomplish the latter, I initially reached out to Krista Voth, a contact with the Vancouver Parks Board. She, and her colleagues, supported our harvesting of bitter cherry bark and canary grass by sharing maps of where these resources grew within city parks. They walked those parks to ensure we had good directions, sent us maps, and provided us with complimentary parking passes. They also let park attendants know we would be there, so that we would not be questioned or feel unwelcome. We were able to harvest in both Jericho and Stanley parks with their support. Both are situated in highly urban settings.

We did not have the same experience trying to connect with folks at the provincial parks to gain access to Cypress and Seymour mountains for our sapling wood and cedar roots. While my Indigenous colleagues, from the host nations, were comfortable with the idea of asserting their rights to harvest within their traditional territory, I had to consider the legal implications for myself and our film maker—Calder Cheverie. In the end, I sought to carry out all our gathering activities with park consent. This eventually led me to drop by the Ranger Station at the Lower Seymour Conservation Area, one week before our final harvesting excursion, to connect with staff from the Metro Vancouver Watershed Management team. They also accommodated our request for access, on short notice, providing us with access to the old growth forest and listening to our needs by opening different areas to us when we were not finding what we needed. One of their staff

even guided us into remote areas not usually accessible to the public and left us with a handheld radio to call for assistance should we need it, allowing us to work in private.

During the project, Gay Williams, the knowledge holder who taught us, was joined on many occasions by extended family members—her daughter Pilasi, her sister Louise, her mother, and three grandchildren. Members of the curatorial collective included me and Jasmine Wilson (of Musqueam) representing the Museum of Vancouver; Carleen Thomas, Jason Leeson, and Michelle George for Tsleil-Waututh Nation; and Chantal Newman, Leateeqwhia Daniels, and Tracy Williams (with Tyselle Newman and occasionally other summer interns) from the Squamish Nation Language and Culture Team. On one occasion we were joined by Sandra Guerin of Musqueam. Our work was filmed by an outdoor educator and film maker, Calder Cheverie, who had previously worked with the Squamish and Tsleil-Waututh nations.

The members of the curatorial collective are our regular cultural liaisons with these nations. Workshops were offered over multiple days to ensure that as many people as possible could attend each type of activity. Filming ensured not only that there would be materials to generate teaching curricula, but that there would be a record for those who missed an event. Project funding was not only available for language translation work, but for the time that these key staff members were spending attending Knowledge Repatriation meetings and workshops. When working with artists and other community members, the museum pays CARFAC fees or honorarium.[9] When engaging with staff from the local First Nations, it pays fees to the respective nations for project review and feedback and the cultural support work that follows. This recognizes the capacity that is being dedicated to our partnership by each nation.

### 3.1. Decorative Materials: Bitter Cherry Bark and Canary Grass

The Knowledge Repatriation Project was organized into four sets of workshops that were held between April 2022 and February 2023. The first two were for acquiring bark and grass needed for creating designs on the baskets. Each of these materials is processed into ribbon-like spools or bundles. These materials are later sewn into place or folded into the basketry coils using a method called imbrication. For the first workshop, we went to Vancouver's Jericho Park in late April when the trees were in flower. Along the Pacific Northwest, many barks are harvested in the spring as this is the time of year where it causes less stress to the tree, and the presence of tree saps makes it easier to remove the bark. We were harvesting bitter cherry bark, which is used for the red and black elements of coiled basketry designs. We went for two consecutive days with different community members joining us on each of the days. We harvested from different types of growing conditions. We offered tobacco when we carried out this work.

The bark that was removed on the first day came from a grove of larger trees, and it was rolled backwards to flatten it and tied in bundles to be dried. Only one tree was selected for harvesting as there were only three in the grove. On the second day, we harvested from smaller trees and in some instances smaller seedlings were removed to support the growth of the other cherry trees. We learned to remove bark in a ribbon from these smaller trunks (Figures 1 and 2).

Our second harvesting expedition occurred in June and was for canary grass, which is used to create the white elements in basketry designs. This was a trickier activity to schedule as we needed to harvest the grass before it went to bloom (if this happened, Gay shared that it would change the texture, and it would not be usable). Readying the grass for use was a four-day activity—one that required it to dry on a line in the sun after it was prepared. Harvesting of the grass went very quickly, requiring only one morning's work to acquire several bundles. Our group was quite pleased with our harvest, and Gay laughed good naturedly, telling us that our combined efforts were typical of "just one of the old ladies" from her community.

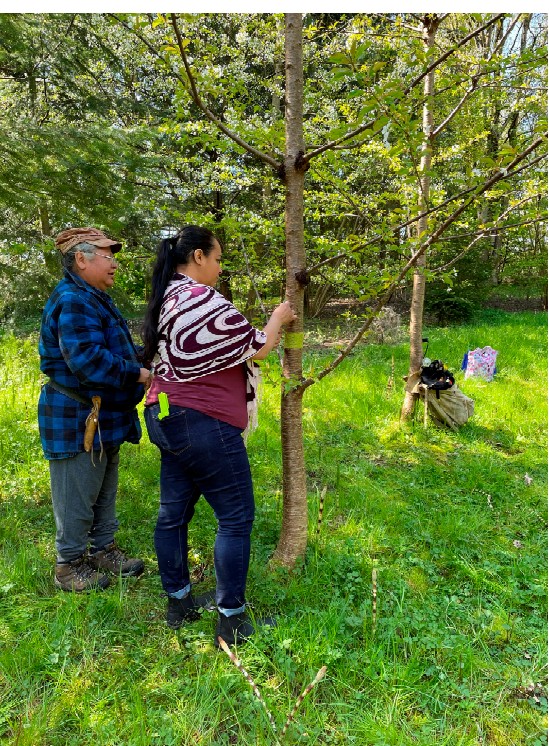

**Figure 1.** Gay Williams of Lil'wat teaches Chantal Newman of the Squamish Nation to harvest bitter cherry bark, 28 April 2022. Photo by Sharon Fortney.

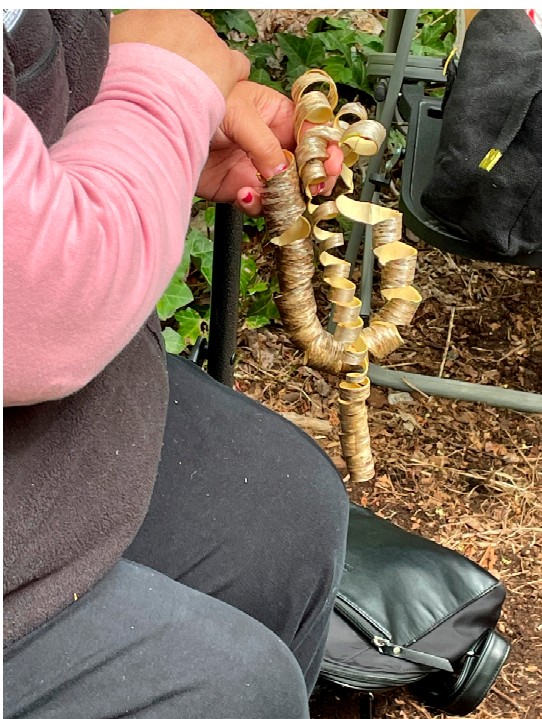

**Figure 2.** A ribbon of bitter cherry bark, 29 April 2022. Photo by Sharon Fortney.

Once we had our grass, we needed to prepare it. In an outdoor space, near the Squamish Nation offices of the Language and Culture Team, we boiled a large pot of water. We placed all the grass into a large plastic fish cooler and covered it with sacking and dish clothes, weighting them down with stones. Boiling water was poured into the cooler to steam the grass, and the lid was closed for an hour before we drained the water out. The

bundles were meant to hang in the sun for four days after this event, to allow them to change color from green to pale yellow (Figures 3 and 4).

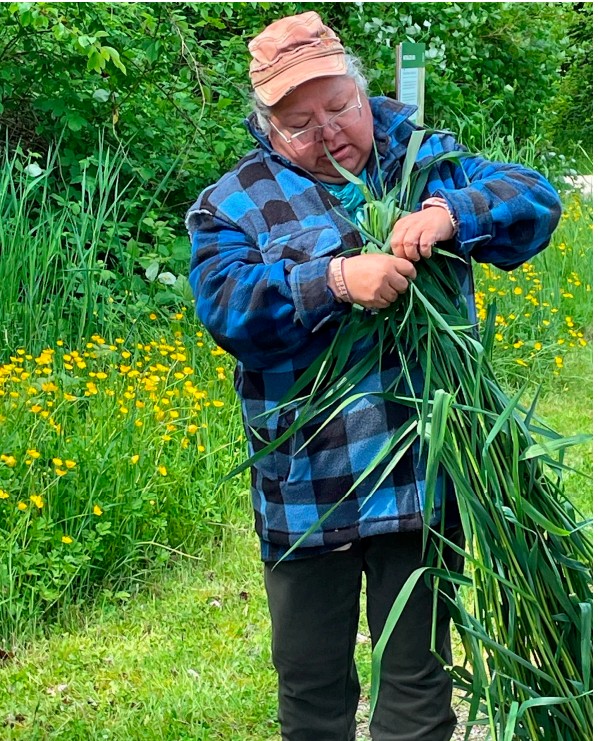

**Figure 3.** Gay Williams demonstrates how to bundle canary grass for transport, 6 June 2022. Photo by Sharon Fortney.

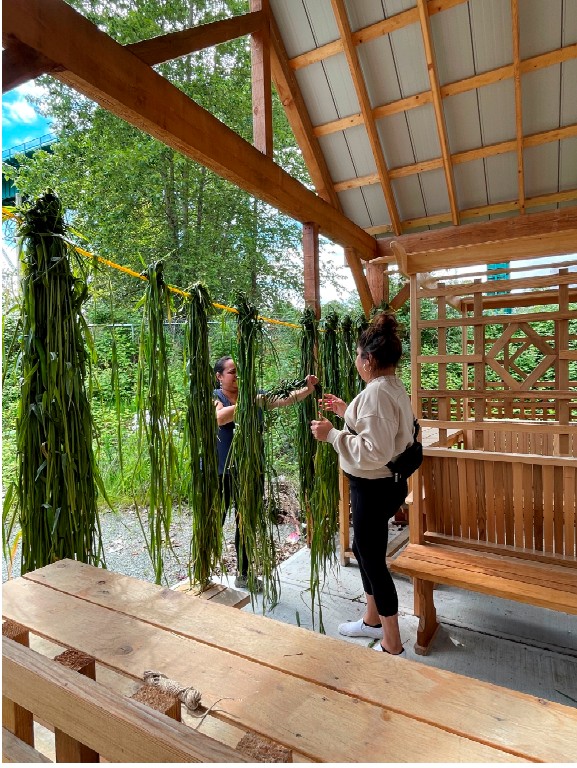

**Figure 4.** Hanging the canary grass after it has been steamed to bleach in the sunlight for four days. Photo by Sharon Fortney.

Unfortunately, rain was in the forecast, but we were able to hang the grass in a covered picnic area, in sunlight but under a roof. After four days, we returned to remove the outer leaves, not needed for our work. All unused materials were carefully collected throughout our project, and those that could not be used in another way were given back to the forest by participants.

### 3.2. Basketry Materials: Sapling Wood and Cedar Roots

I have always been taught that you gather your cedar roots in the spring when the sap is running in the trees. During this project, I learned that this differs in other areas of the Salish world. When logistical concerns prevented us from scheduling our root harvesting in the summer, Gay suggested we completed it in the fall. What we learned from working with Gay was to pay attention to the growing conditions. We also needed different sizes of trees for sapling wood versus cedar roots.

The first day we harvested cedar roots we had little success as we were in an area where the trees had previously been logged and there was little undergrowth. Conditions were also drier. We found hemlock roots growing interspersed and had to pay extra attention to the texture of the roots so as not to confuse them. Gay noted that, in recent years, the inner bark of the hemlock has changed to have a reddish appearance that was more typical of cedar roots. A few times this tricked us into harvesting the wrong roots. This color change led us to speculate on the impacts of climate change on local trees. We tried a second location where the trees were not large enough for root gathering, but were suited to sapling wood, so we changed our plans.

When we went for sapling wood, we wanted younger trees—we paid attention to the diameter of their trunks and looked for trees that had a surface with fewer limbs. The harvesting was performed in a manner similar to the harvesting of cedar bark, except we cut deeper with hatchets and used wedges and ropes as part of the live harvesting. This was a physically harder task than harvesting cedar bark as well, and we at times were lined up on the rope like a tug-of-war team. After harvesting, we spent time sitting together, splitting our wood into slats and bundling them for later use (Figure 5).

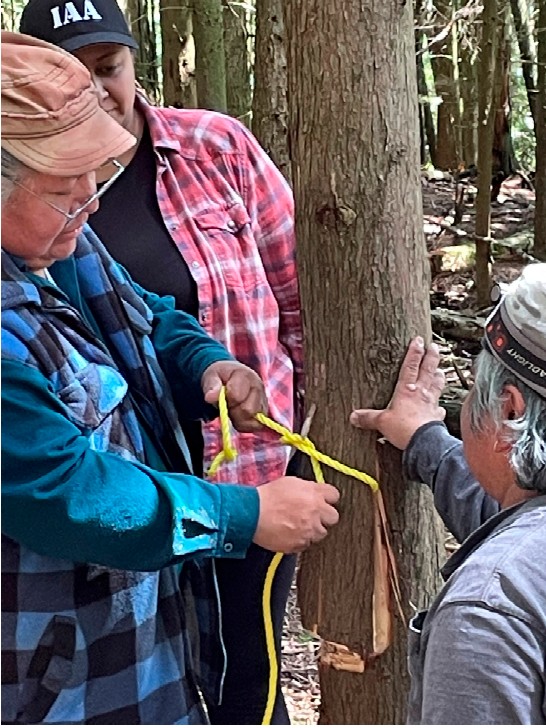

**Figure 5.** Gay Williams and her sister Louise demonstrate how to live harvest cedar sapling wood. Photo by Sharon Fortney.

On our second day of root harvesting, we were escorted several kilometers into the conservation area by Dayna Timmons from Metro Vancouver's Watershed and Environment Management Team to an old growth region located below the watershed. This was a conservation area that is not easily accessed. We were looking for large cedar trees with mossy logs lying beneath them. Although we found some, they were on a rocky slope. We looked inside the fallen logs for roots that were large and straight enough, but those we found were drier than preferred, and the area was too rocky, making the roots kinky. Despite the trees being the right size, Gay proclaimed the conditions not ideal.

On the third day, Gay and Pilasi scouted along the road to the conservation area. They were looking for groves of larger cedars interspersed with broadleaf maples and undergrowth such as salal and ferns. Ironically, they found a small grove that was ideal adjacent to the visitor parking area, so we proceeded to harvest there at the entrance to the conservation area. After harvesting our roots, starting a few feet away from the base of the tree, we notched the top of the root and removed the outer bark and rootlets. We later used the notches for splitting the roots into withes. These roots, gathered on our third day, had a moistness to them that made removing the outer bark an easy task. They were bundled and set aside for later use.

### 3.3. Making a Cedar Root Basket

Harvesting workshops were designed to be flexible for attendance, but the basket making one needed to work for everyone's schedule as we only had enough funds to offer it once. After determining everyone's availability, I asked Gay to select a project that we would be able to accomplish in two days. Gay decided that we would learn to make a small basketry canoe as our first project. She selected this project because it involved all the techniques needed to make a basketry cradle, and those found in other types of baskets. We gathered in February 2023 for a two-day workshop at the Museum of Vancouver. The first day was largely spent making our roots ready to work with. We split them down if they were too thick and smoothed them under our knife blades, trying to remove knots. We cut our sapling wood into 8-inch lengths and thinned them at the ends. We selected six of the same width, and two longer ones, for our project. The longer ones were used to hold the slats in place while we twined cedar roots around them, as would be performed to create the base of a basket. They were later removed.

Once our slats were completely covered with root, we stitched the ends to secure them. This created rows of x's along two edges. We then folded our work in half, so there were three slats on each side, and using our awls sewed the ends together. This was a very hard task. When it was complete, we cut a little "bench" for our canoes from a piece of sapling wood and inserted it near the center top. This helped form the shape of our canoe. We then gathered a bundle of less-desirable roots, those that were too thin or uneven, and secured them in place on the rim with one or two stitches. We then used awls to sew the top coil in place and finish the basket. This task was much easier than sewing the ends of the canoe together. The latter required sewing through two cedar slats, and we learned that this was why it was important to thin the ends. A small slat foot was then added to the base to help our canoes stand upright. This small project took us two days, leaving none of us beginners with time for adding decorative materials (Figures 6 and 7).

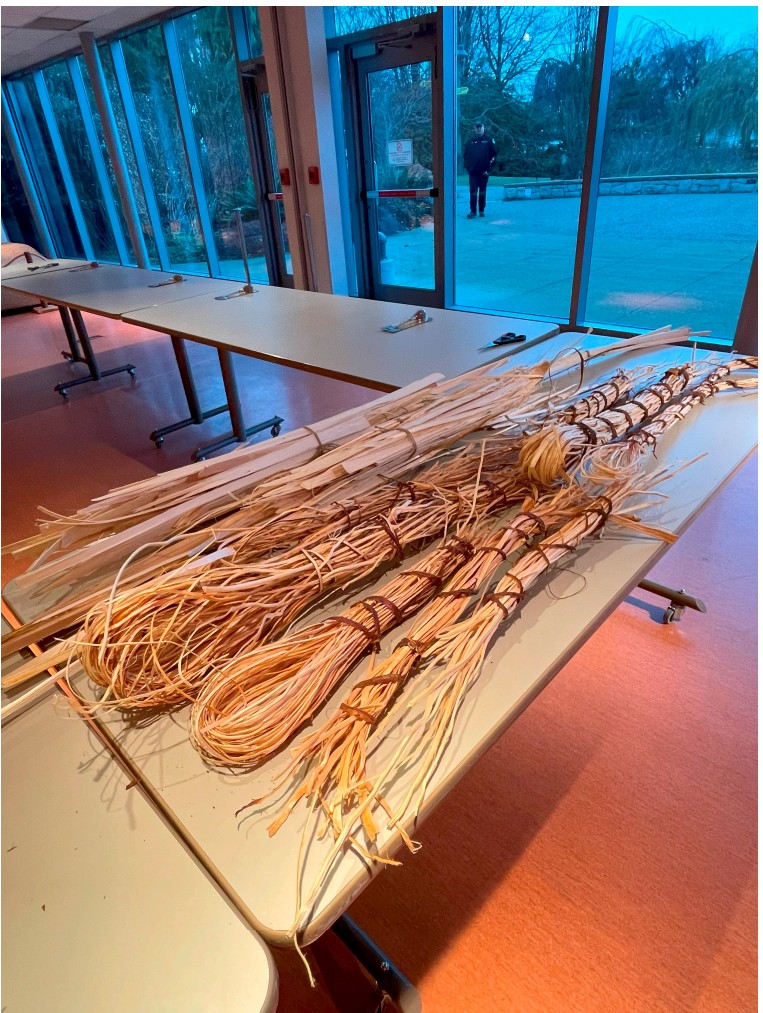

**Figure 6.** Bundles of cedar roots at the Knowledge Repatriation Basketry Workshop on 16 February 2023. Photo by Sharon Fortney.

As this project is intended to build capacity within the participating nations, by creating opportunities for those who teach language and culture, and facilitate opportunities for others in their communities, all the activities I have described were filmed and photographed. We are creating four short films as curriculum materials for use within the host nation communities. We intend to add scene breaks in each film, featuring the two local languages, but our initial plan to completely translate the films was reduced in scope when we recognized there was not enough capacity to accomplish this task. When working with endangered languages, it may often require many hours of research to recover missing vocabulary or find agreement on the best way to express an idea. This sometimes involves consulting those who work with neighboring dialects or other Salish languages to investigate what terminology they have documented. At the end of the project, we will retain the raw footage at the museum as the communities may wish to use it to inform a future exhibition or program. Each of the participating communities, and Gay, will also receive the raw footage and films for their archives after the films have been edited.

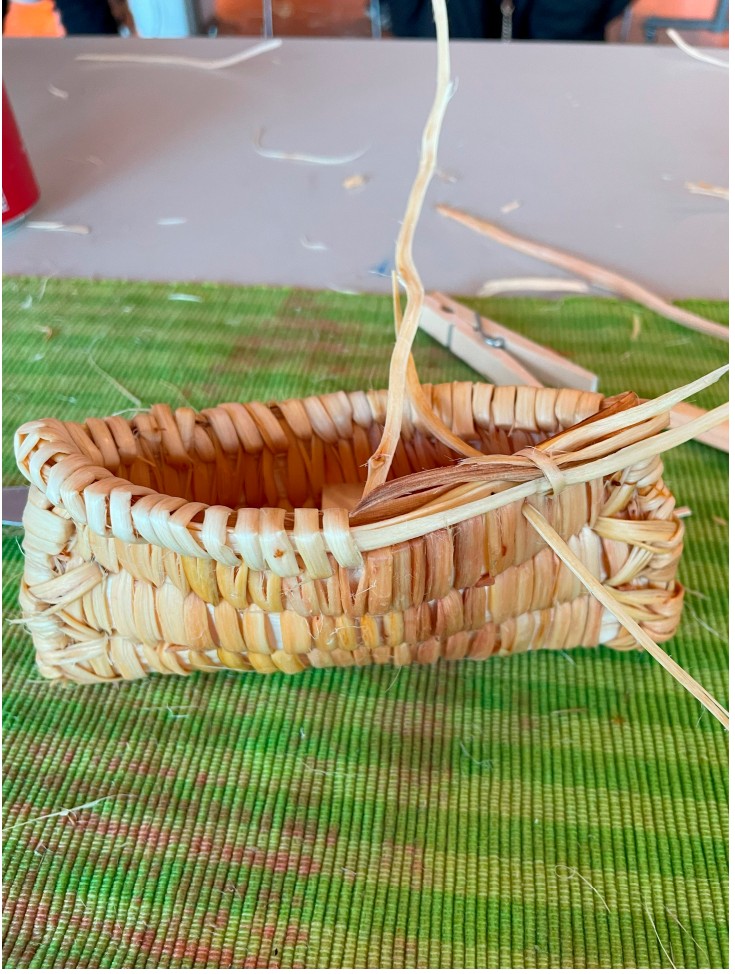

**Figure 7.** A small canoe basket in progress. Photo by Sharon Fortney.

### 4. Final Remarks

My description of the work we carried out is very cursory. It is hard to capture several days' worth of teaching in a few paragraphs. What I hope this article conveys is the complexity of knowledge that goes into creating belongings by Indigenous community members. How knowledge is shared can be family-specific and highly variable. That knowledge is also linked to distinct places on the land. It frequently requires experience to understand the right conditions for gathering and creating, and some of it may be determined by seasonal cycles.

Initially, this project was about making coiled cedar root baskets as they are no longer produced in the Greater Vancouver area. It was also about rediscovering the types of places that produce the plants needed for this work and learning to identify the right conditions for harvesting. Gay was always teaching us, and some of her stories involved young girls in her community learning to make baskets and putting them out along a trail for those who needed them to take home. It is important to pay respect to the plants and trees that we harvest, and to give away our first work as a sign of respect and to avoid being stingy. Samples of materials from our harvests, and my first root basket, were added to the MOV collection in case they are needed for a future exhibition or program. They join older cedar roots gathered from the Toba Inlet—where my grandmother lived as a girl—and samples of barks and grasses collected from the Spuzzum community by a basket maker named Annie York. Today, these types of materials are of great interest to contemporary basket makers and other emerging artists.

As we reshape the Indigenous collections at the museum through repatriation work and contemporary collecting, we also reshape the focus from provincial to local, recognizing

that our museum is about the people of "Vancouver" and their stories. Many stories could be told by the work we carried out on this first knowledge repatriation project—some of the themes speak to sustainability and environment, others to reconciliation and redress, while the act of harvesting and creating was also one of reclaiming memory and identity.

While we achieved the identified objective of making small, coiled cedar root baskets, this project was ultimately about creating working relationships that are not extractive. This means continuing to explore ways that the museum can support capacity building and cultural renewal in the three host nation communities. At the time of writing, planning has begun for a second year of the Knowledge Repatriation Project. The focus has shifted to sharing knowledge across the Salish Sea, at the request of Squamish and Tsleil-Waututh representatives, and funds have been secured to undertake two excursions to visit knowledge holders. The first workshop was taught by Suquamish Elder and Master Weaver Ed Carriere at his home in Indianola (Figure 8). We learned how to make basketry shrimp and fish traps and work with stinging nettle fibers to make duck nets. We also harvested tule reeds to make mats. The second visit will be to sea gardens located in the Gulf Islands to learn about maintaining clam beds. Parks Canada is helping us to connect with W̱SÁNEĆ and Hul'q'umi'num knowledge holders to learn about how they care for their clam gardens in partnership with the parks staff. This latter project is of special interest to the Tsleil-Waututh community, as they have been working to restore the waters and shorelines of their traditional territory and now have one productive shellfish beach. For many decades, pollution has prevented them from harvesting clams and cockles—shellfish harvesting completely ceased along the shores of Eastern Burrard Inlet in the 1960s (Morin 2015, p. 339).

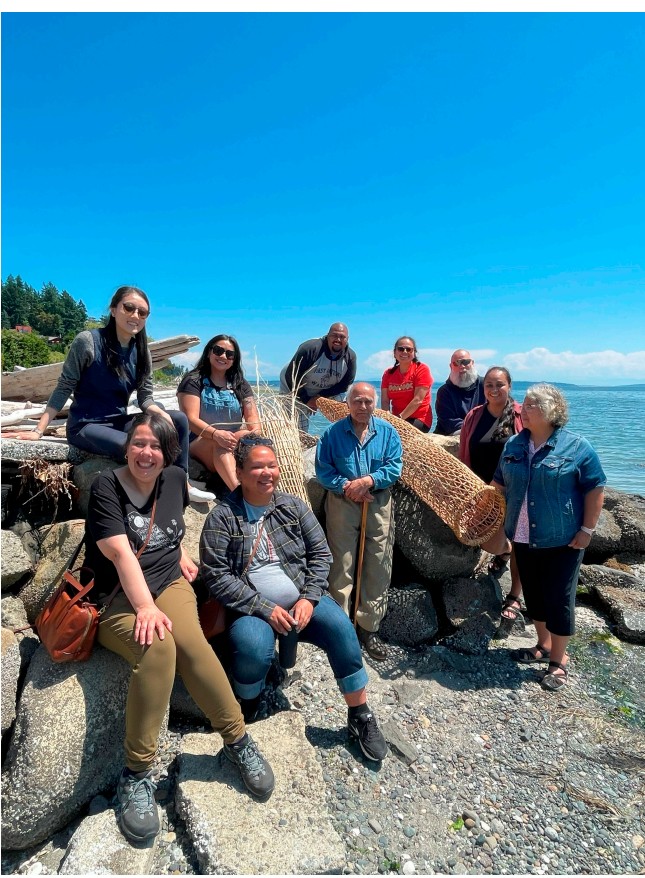

**Figure 8.** Visiting with Suquamish Elder Ed Carriere in Indianola, Washington, to learn about basketry fish traps, 27 June 2023. Front row (L–R): Sharon Fortney, Michelle George, Ed Carrier, Chantal Newman, Carleen Thomas. Back Row (L–R): Charlotte Chang, Jasmine Wilson, Willie Lewis, Leateeqwhia Daniels, and Jason Leeson. Photo by Sharon Fortney.



The work we are carrying out around knowledge repatriation is only possible when work is also being carried out to support environmental stewardship, and the restoration and protection of plant and animal resources. Access to materials is as much a part of the process as having the knowledge of how to construct an object. When we look at older belongings in museum collections, and the materials they are made from, we see how our ancestors moved about on the land and water of their territories and we have an opportunity to gain insights into the seasonality of their movements. Belongings are not just functional objects, they represent knowledge about lands and resources, and by learning to make them we also learn important knowledge about stewardship and leaving a legacy for future generations. There is a web of knowledge that surrounds all of our material things and by rediscovering one aspect, we gain access to much, much more.

**Funding:** Funding received from Vancouver Port Authority for harvesting activities. Canada Council of the Arts funding was used for filming. The next phase of the project is being funded by the Department of Canadian Heritage and BC Arts Council.

**Data Availability Statement:** The data generated by this project will be archived at the Museum of Vancouver, Musqueam Indian Band Archives, Squamish Nation Archives, and Tsleil Waututh Nation archives.

**Conflicts of Interest:** The author declares no conflict of interest.

## Notes

1.  Amendments to Canada's Indian Act in 1884 included the Potlatch Ban and the Indian Residential School Act. The Potlatch Ban, which prevented Indigenous people from gathering for ceremony, was in effect until 1951. The last federally operated Residential School closed in 1996 in the province of Saskatchewan.

2.  Staff at the Museum of Vancouver, and our neighbors at the UBC Museum of Anthropology, have adopted the use of the term "belongings" to describe the Indigenous collections we care for. This move was led by the Musqueam community during a joint exhibition project about the ancestral village of c̓əsnaʔəm. Community members told museum staff "these are not artifacts they are our ancestors' belongings".

3.  On another occasion I was told that the community of Musqueam had received 165 requests for protocol in a single week. The 2016 census notes a population of 1652 community members, 26% of which were under the age of 19.

4.  Visit https://vancouver.ca/people-programs/plaza-naming-project.aspx accessed on 10 March 2023, to learn more about the renaming of civic plazas by the City of Vancouver, including how to pronounce the new names.

5.  Similar work has also been carried out by the Musqueam community at UBC, where the malls of the University have been given new place names as an act of reterritorializing the campus adjacent to the main community located at Musqueam IR2. Visit Musqueam Street Signs | UBC Campus & Community Planning.

6.  For storybooks, visit Stories | Musqueam Teaching Resource (ubc.ca).

7.  For examples of baskets from my family visit the online exhibit "Honouring the Weavers: North Coast Salish Baskets and Basket Makers" (https://baskets.crmuseum.ca/weaver/annie-pielle-chapman) accessed on 10 March 2023.

8.  Irene and I were both employed as basketry researchers for the Canadian Museum of Civilization curator, Andrea Laforet, in 1999.

9.  CARFAC fees are the national standard for artists' fees in Canada, including exhibition, reproduction, and other services. They are updated each year.

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
