# Peer review of "Knowledge Repatriation: A Pilot Project about Making Cedar Root Baskets"

_arts, 2023_

Round 1

Reviewer 1 Report

This is a very significant article that contributes Indigenous perspectives and Indigenous knowledges to a field of study in museology, Indigenous culture and knowledge, and Indigenous fibre art. It is very well researched, particularly in terms of practice-led research, and is very well written. 

I have two further comments that are more suggestions for further consideration rather than requirements for changes. The first is that the dichotomy of 'extraction' purposes versus contribution to Indigenous communities purposes possibly needs not be so definitive. Certainly the key concern here is that the research is driven and conducted by Indigenous people from host communities. However whilst these projects support language regeneration and cultural regeneration and regaining knowledge in the communities, it is at the same time beneficial for the broader community that this knowledge is being regenerated. It is a part of global cultural heritage that matters to everyone even if it is not necessarily immediately accessible to everyone. Some time in the future this knowledge may be shared with outsiders through exhibitions, but just because the primary immediate issue is for the host communities, it does not mean that it is not globally beneficial as well. 

The second point is that I have a bit of a concern that possibly too much information is being shared about the where and how of harvesting. I know that in Australia where there is a lot of Indigenous weaving generation occurring, we are seeing quite a few skilled non-Indigenous weavers trying to learn the where and how of harvesting so that they can create weavings using the Indigenous techniques. I understand that the article does not explain how to make the baskets explicitly, but for people with skills in similar techniques, the knowledge of how and where to collect the materials has often incited them to go to these places and take the reeds, bark, etc., without Indigenous approval. Often this means that too much is taken, or at the wrong time. Reeds ready for harvest have even been crushed under foot necessarily and the year's harvest is ruined. I am wondering if the current article could still convey the complexity of knowledge that is invested in these cultural practices without providing as much specific knowledge about where and how to harvest?

I enjoyed reading this article very much and appreciate that it has great value in conveying the degree of knowledge and holistic consciousness that is invested in such regeneration practices, and their relationship to language regeneration for host communities. 

Author Response

Thank you for your thoughtful review. I agree that there is some risk that non-Indigenous artists might consider how they can harvest and reproduce some of this knowledge. I have already had many inquiries from Indigenous community members, from different communities in other parts of Canada who would like to participate, which I have had to politely ignore. I will consider ways to introduce these ideas into the text as I do my revisions

Reviewer 2 Report

A very important document to assist in non-Native understanding of tribal culture and history. Thank you for bringing this to the forefront. (BTW: I am Powhatan).

Author Response

Thank you for reading my article and your words at the end. I am happy to be sharing this work and I appreciate your positive feedback.

Reviewer 3 Report

Overall, this article is a valuable model for collaboration between mainstream museums and First Nations communities. Readers familiar with other First Nations/Indigenous communities will find the article relatable and that the author's community-based perspective is useful for making sense of the challenges such collaborative projects face.

Specific areas that would benefit from further development are noted below.

The article raises important points about the need for knowledge repatriation, but the argument could be better supported in the Conclusion. Also, there is no reference to scholarship on museum collaboration, such as the work of Amy Lonetree. Academic audiences would find it useful to know how this author's analysis relates to or departs from other studies of museum/community collaboration.

Page 2: The author brings up important point about how the Squamish receive an inordinate amount of requests. This is something that other tribes can relate to and is worthy of expansion. Why is this point so important? I would encourage the author to expand and explain why requests from outside place a heavy burden on First Nations/Native American tribes with limited resources and community representatives who often work full-time jobs. This issue could be tied to the larger issue of museums needing to be sensitive to the needs and limitations of Native communities.

Page 3: This is the first place where the author identifies themself as Coast Salish and being from a basket weaving family. This should be introduced at the beginning of the article so the reader has a clear understanding that the author is providing an Indigenous perspective and why this work is so important for the author.

Pages 4-5: When discussing the gathering trips to the city and provincial parks, it would help the reader if the author gave a sense of when this occurred. Year or month and year? Also, was season a factor in determining when they gathered? See comment below.

Page 5: What is CARFAC? Can the author spell this out?

Page 5: Tied to question above: At the bottom of the page, the author states that they harvested bitter cherry bark in April. Could the timing be specified at the first mention of Jericho Park? Why did they gather in April? Do trees need to be in flower in order to gather bark for weaving?

Page 6: Good explanation as to why June was the month they chose to gather.

Page 7: Author explains the times of year when basket makers gather materials. This explanation should appear earlier in the article, before the discussion of the gathering trips. Overall, the article would benefit from some re-organization.

Page 10: There is an abrupt shift from the first to the second paragraph. A transition would help lead the reader from the discussion of the basket project to the filming. A discussion of how basket makers tend to work together, for instance, would give the reader a better sense of why making the baskets as a group has cultural significance even though they were also factoring in budgetary constraints.

Page 12: The author refers to objects in museums as belongings here and in the Introduction. Can they explain why they chose to use this word? Why is “belonging” important in a study of knowledge repatriation? The argument could be strengthened if the author fleshed out the concept of “belongings” and included references to other scholars/cultural practitioners who use this term.

Author Response

Thank you so much for your review of my article. I have made many of your changes, and clarifications. I appreciate the attention you gave to everything.